# Effects and Mechanism of the *Leontopodium alpinum* Callus Culture Extract on Blue Light Damage in Human Foreskin Fibroblasts

**DOI:** 10.3390/molecules28052172

**Published:** 2023-02-26

**Authors:** Xianyao Meng, Miaomiao Guo, Zaijun Geng, Ziqiang Wang, Huirong Zhang, Sunhua Li, Xiao Ling, Li Li

**Affiliations:** 1College of Chemistry and Materials Engineering, Beijing Technology & Business University, Beijing 100048, China; 2Acelbio (Chongqing) Biotechnology Co., Ltd., Chongqing 404100, China; 3Beijing Lan Divine Technology Co., Ltd., Beijing 100872, China

**Keywords:** blue light, *Leontopodium alpinum* callus culture extract, opsin 3, collagen, metalloproteinase 1, calcium influx, reactive oxygen species, leontopodic acid A, leontopodic acid B

## Abstract

*Leontopodium alpinum* is an important source of raw material for food, medicine, and modern cosmetics. The purpose of this study was to develop a new application for protection against blue light damage. To investigate the effects and mechanism of action of *Leontopodium alpinum* callus culture extract (LACCE) on blue light damage, a blue-light-induced human foreskin fibroblast damage model was established. The contents of collagen (COL-I), matrix metalloproteinase 1 (MMP-1), and opsin 3 (OPN3) were detected using enzyme-linked immunosorbent assays and Western blotting. The calcium influx and reactive oxygen species (ROS) levels were measured via flow cytometry and the results showed that the LACCE (10–15 mg/mL) promoted the production of COL-I, inhibited the secretion of MMP-1, OPN3, ROS and calcium influx, and may play a role in inhibiting the activation of blue light on the OPN3-calcium pathway. Thereafter, high-performance liquid chromatography and ultra-performance liquid chromatography–tandem mass spectrometry were used to quantitatively analyze the contents of nine active ingredients in the LACCE. The results indicated that LACCE has an anti-blue-light-damage effect and provides theoretical support for the development of new raw materials in the natural food, medicine, and skin care industries.

## 1. Introduction

Blue light is visible light with a wavelength of 400–2500 nm, also known as high-energy visible light [1], which is the band with the shortest wavelength and highest energy in the visible spectrum [2]. The main sources of blue light are sunlight and electronic screens; although luminous LED screens and fluorescent lamps can also serve as additional sources of blue light [3]. Photoaging consists of decreased skin elasticity in rhytids, lentigines, and mottled pigmentation. Blue light has been shown to induce photoaging in vitro and in vivo, causing oxidative stress, damaging the skin barrier, and promoting persistent skin pigmentation, among other effects. Previous studies have shown [4,5,6] that LED-BL-irradiated skin cells (keratinocytes and fibroblasts) induce reactive oxygen species (ROS) production and DNA damage, resulting in an increase in metalloproteinase 1 (MMP-1, collagenase) and a decrease in collagen (COL-I). However, one study [7] shows that the intensity of blue light emitted by electronic devices is not as harmful to skin as sunlight. The blue light that affects the skin comes mainly from sunlight, and the truly harmful blue light is in the range of 400 to 450 nm [8]. The damage to skin caused by electronic devices requires further study of more parameters besides wavelength, such as total dose, irradiance and chronic exposure time, as well as skin type. Furthermore, after exposure to blue light, people with dark skin (type III and above) will develop long-lasting and intense pigmentation [9].

The effect of blue light in causing damage depends on different photoreceptors; the main and most important being opsin (OPN), flavoproteins, porphyrin-containing enzymes, and nitrosated proteins. Currently, OPNs are receiving increasing attention as potential targets of phototransduction. They are a series of light-sensitive G-protein-coupled receptors (GPCRs) [10,11]. Specifically, OPN photoreceptors absorb specific wavelengths and are responsible for the transduction of visible light. However, non-imaging forming OPNs have been identified in the skin. Skin OPNs can regulate circadian rhythms, epidermal barrier function, and melanogenesis. For example, BL activation of OPN3 may increase keratinocyte differentiation and tyrosinase activity in melanocytes [12]. OPN3 is most abundant in the skin [13]. 

In particular, OPN3 is an important membrane protein that transduces extracellular signals and has been confirmed to be significantly associated with pigmentation in human epidermal melanocytes [14], is essential to restore barrier functions [15], and is an important target protein mediating blue light transduction. Regazzetti et al. [14] determined that blue-light-induced pigmentation in darker skin types results from the activation of OPN3, which activates pathways such as extracellular signal-regulated kinase and p38, leading to the phosphorylation of the microphthalmia-associated transcription factor (MITF), tyrosinase activity, and subsequent melanogenesis in the melanocytes. Lan et al. [16] demonstrated that OPN3 acts as a sensor for UVA phototransduction in fibroblasts and participates in the upregulation of MMP-1 and MMP-3 expressions, and clarified that UVA leads to the phosphorylation of downstream mitogen-activated protein kinase (MAPK)/activator protein-1 (AP-1) signal-related proteins through the OPN3-calcium-dependent signal transduction pathway. After the activation of the MAPK pathway, transcription factors c-Fos and c-Jun become heterozygotic to form AP-1, which further regulates the synthesis of collagen and matrix metalloproteinases, finally upregulating the expression of MMP-1 and MMP-3 and inhibiting the expression of COL-I. The effect of blue light on photoaging is similar to that induced by UVR. Blue light irradiation can also significantly increase the ROS expression level of fibroblasts and activate the c-Jun N-terminal kinase (JNK)/AP-1 signaling pathway [17]. The MAPK/JNK/AP-1/COL-I pathway is one of the molecular mechanisms through which blue light inhibits collagen synthesis in cells, which is also consistent with the mechanism of UV-induced skin aging.

*L. alpinum* is a perennial herb from the family Asteraceae, commonly known as edelweiss, alpine snow grass, and a small white flower. It is native to the Alps mountains in Europe and grows in high-altitude limestone areas. It is also the national flower of Switzerland and Austria [18]. *L. alpinum* is known for its unique medicinal properties and is highly self-healing, resistant to severe weather [18]. It has bacteriostatic [19], antioxidant [20], blood-lipid-regulating [21], anti-inflammatory [22], analgesic [23] and other pharmacological effects. Its chemical components are mainly volatile oils, flavonoids, phenylpropanoids, sesquiterpenoids, steroids, and other compounds, including phenylethanoid glycosides, which have significant antioxidant, anti-inflammatory, and anti-aging effects [24]. However, due to its harsh growing environment, scarce resources, and increasing human demand, the species is now considered rare. Therefore, the commercial availability of this plant for industrial use is promoted with cell tissue culture technology for the regeneration of edelweiss plants and the protection of rare and endangered plants. 

*L. alpinum* extract is well known for its use in pharmaceutical preparations, but its use in the skin care industry has been poorly reported. Therefore, in this study, LACCE was extracted from the callus of *L. alpinum*. The antagonistic activity of LACCE against blue light and its mechanism in human foreskin fibroblasts (HFF) cells were studied using enzyme-linked immunosorbent assays (ELISA), flow cytometry, and Western blotting (WB) methods, and the LACCE components were quantitatively detected via ultra-performance liquid chromatography–tandem mass spectrometry (UPLC-MS/MS) and high-performance liquid chromatography (HPLC). In this regard, *Leontopodium alpinum* callus culture extract (LACCE) can protect the skin from some of the damage caused by blue light. Our results provide a theoretical basis for expanding its application to anti-blue light damage. 

## 2. Results 

### 2.1. Analysis of the LACCE Chemical Composition

In this study, seven components, namely syringin, chlorogenic acid, cynarin, isochlorogenic acid A, asiatica, isoquercitrin, and isochlorogenic acid C, were quantitatively analyzed using HPLC (Figure 1, Table 1). The active components of leontopodic acid A and B were quantitatively analyzed using UPLC-MS/MS (Figure 2, Table 2). The regression equation is the average of the three curves.

### 2.2. High-Energy Visible-Induced Photodamage Model in HFF Cells

#### 2.2.1. Effect of Blue Light on HFF Cell Viability

First, the effects of different doses of blue light irradiation on the viability of HFF cells were measured using the cell counting kit-8 (CCK-8) assay (Figure 3). When the blue light dose was 0, 6, 12, or 18 J/cm^2^, the cell survival rate was >80%. Still, cell survival rate decreased with increasing blue light intensity. Thereafter, ELISA and WB were carried out, and the three doses were detected using COL-I and MMP-1 indexes to establish a blue-light-damage cell model.

#### 2.2.2. Effect of Blue Light on COL-I and MMP-1

The ELISA and WB results showed that when the blue light irradiation dose was 18 J/cm^2^, compared to the 0 J/cm^2^ group, COL-I levels significantly decreased (*p* < 0.05), and MMP-1 levels significantly increased (*p* < 0.05; Figure 4). Therefore, 18 J/cm^2^ was considered the best dose to establish a model of blue light damage in HFF cells.

### 2.3. HFF Cell Viability after LACCE Treatment

The effects of different concentrations of LACCE on the viability of HFF cells were measured using the cell counting kit-8 (CCK-8) assay (Figure 5). When the concentration reached 20 mg/mL, the cell survival rate was less than 80%. Therefore, four LACCE concentrations with a cell survival rate of more than 80% (≤15 mg/mL) were selected for the next experiment.

### 2.4. Effects of the LACCE on COL-I and MMP-1 Levels

Compared to the control group, the COL-I levels significantly decreased and MMP-1 levels significantly increased in the blue light model group (*p* < 0.01). Moreover, compared to the blue light model group, the LACCE significantly promoted COL-I secretion and significantly inhibited MMP-1 secretion in the HFF cells, at a 10–15 mg/mL concentration (Figure 6 and Figure 7). 

### 2.5. Effects of the LACCE on the OPN3-Calcium-Dependent Signal Transduction Pathways

#### 2.5.1. Effect of the LACCE on OPN3 Levels

Compared to the blank control group, OPN3 levels in the blue light model group significantly increased (*p* < 0.01). Compared to the blue light model group, the LACCE significantly inhibited OPN3 levels in the HFF cells at 5–15 mg/mL (*p* < 0.01; Figure 8). 

#### 2.5.2. Effect of the LACCE on Ca^2+^ Flow into the HFF Cells

Calcium ion inflow was detected using flow cytometry. Compared to the blank control group, the Ca^2+^ inflow in the blue light model group significantly increased (*p* < 0.01). Compared to the blue light model group, the LACCE (5–15 mg/mL) significantly inhibited Ca^2+^ inflow (*p* < 0.01; Figure 9). 

#### 2.5.3. Effect of the LACCE on ROS Levels in HFF Cells

Using flow cytometry, ROS experiments were performed on the LACCE-treated HFF cell samples. Compared to the blank control group, ROS levels in the blue light model group significantly increased (*p* < 0.01). Compared to the blue light model group, 5–15 mg/mL LACCE significantly reduced ROS levels in the HFF cells (*p* < 0.01; Figure 10). 

## 3. Discussion 

To study the anti-blue-light-damage effect of the LACCE and its mechanism of action, a blue-light-induced HFF cell damage model was established. Blue light at a dose of 18 J/cm^2^ significantly promoted the secretion of MMP-1 and inhibited the production of COL-I. When studying the safe concentration of LACCE on HFF cells, it was found that LACCE promoted cell proliferation at a concentration of 2 mg/mL compared with the control group, so it is expected to be found to promote cell viability. However, with the increase in the concentration of active ingredients, there may be some cytotoxicity to cells, so it is necessary to explore the correct concentration of LACCE to promote the improvement in cell vitality. In this study, the concentration of LACCE was 5–15 mg/mL to study the anti-blue light damage of LACCE. It was found that concentrations of 10–15 mg/mL LACCE significantly inhibited the production of MMP-1 and promoted the secretion of COL-I in HFF cells. This showed a clear anti-blue-light-damage effect on the HFF cells and indicated that the damage was displayed with an increase in MMP-1 and a decrease in COL-I levels. The mechanism of the LACCE in response to blue light damage was further investigated.

An important membrane protein that transduces extracellular signals, OPN3, has been confirmed to be significantly associated with pigmentation in human epidermal melanocytes, essential to restore keratinocyte and fibroblast barrier functions, and an important target protein mediating blue light transduction [14,15]. Ge’s study [17] showed that blue light induced ROS production in HFF cells and was one of the factors causing changes in MMP-1 and COL-I expression. The study [16] by Lan et al. demonstrated that OPN3 acts as a sensor for UVA phototransduction in fibroblasts and participates in the upregulation of MMP-1 and MMP-3 expression. Similar to the mechanism of UV-induced skin aging, this study started with the hypothesis that blue light may induce MMP expression in fibroblasts via OPN3-calcium-dependent signal transduction pathways. Additionally, this study investigated whether the LACCE could inhibit these pathways. This study revealed that the LACCE significantly inhibited OPN3 levels, calcium ion inflow, and ROS levels. Therefore, this also posed the hypothesis that the LACCE could inhibit the activation of OPN3-calcium-dependent signal transduction pathways and the occurrence of ROS oxidative stress, thereby inhibiting the secretion of MMP-1 and producing an anti-blue-light-damage effect.

Blue light may induce skin damage through multiple molecular pathways. Both the TGF-β and MAPK signaling pathways are key for skin collagen synthesis. TGF-β is most closely related to extracellular matrix synthesis and the secretion of cytokines, and the inhibition of the TGF-β signaling pathway inhibits collagen synthesis. The MAPK signal pathway is activated in various stress response situations, and the transcription factors c-Fos and c-Jun become heterozygotic to form AP-1, which further regulates the synthesis of collagen and matrix metalloproteinases. A study has shown that [17] blue light promotes the synthesis of MMP-1 by activating the JNK/c-Jun/AP-1 pathway and inhibiting the TGF-β pathway through the JNK/SMAD7 pathway, thus decreasing the production of COL-I. Therefore, the MAPK and TGF-β pathways are both molecular mechanisms for the blue-light-driven induction of the skin aging phenotype in HFF-1 cells. Whether blue light activates the downstream MAPK and TGF-β pathways through the activation of the OPN3 receptor and whether the LACCE can play the role of anti-blue light damage by inhibiting this pathway still needs to be verified.

Since the early 2010s, *L. alpinum* plant extracts have been recognized as a very valuable source of anti-aging skin compounds by the skin care industry. The main active substances of these extracts are phenylpropanoids such as leontopodic acids [25]. However, few studies have been conducted on the active constituents in the LACCE. Hence, the contents of nine active ingredients were quantitatively analyzed using HPLC and UPLC-MS/MS, namely leontopodic acid A, leontopodic acid B, syringin, chlorogenic acid, cynarin, isochlorogenic acid A, asiatica, isoquercitrin, and isochlorogenic acid C, among which leontopodic acid A and B were the highest concentrated active ingredients (leontopodic acid A: 184.13 ± 6.21 μg/mL, leontopodic acid B: 161.10 ± 3.85 μg/mL). The edelweiss plant has been recognized as a very valuable source of anti-aging compounds due to its antioxidant compounds, namely leontopodic acids and 3,5-caffeoylquinic acid [26]. In this work, the high contents of leontopodic acid A and B in the LACCE, as well as the other active ingredients, provided some evidence for the anti-blue light effect of the LACCE. However, more research is needed to understand the complex active ingredients and biological roles of this plant. Leontopodic acid may be the main ingredient responsible for the function of *L. alpinum*. Thus, further studies could isolate leontopodic acid from the *L. alpinum* callus to further study its effect.

In this study, it was found that the LACCE can inhibit blue light damage by inhibiting OPN3 receptor activation and the ROS oxidative stress response, thereby inhibiting MMP-1 secretion and promoting COL-I synthesis. In the future, the mechanism of blue light on the OPN3 receptor and between the MAPK and TGF-β pathways will be investigated. In addition, whether the LACCE can exhibit its anti-blue-light-damage role through these pathways, as well as the structural activity relationship between the LACCE and its active ingredients, mainly leontopodic acid, will be the focus of further research. Additionally, this study was conducted in cells, not in a full skin model. Skin is a much more complicated and complex system and the observed results may be not the same. Additionally, the skin is affected differently by sunlight than by the blue light emitted by electronic devices. Therefore, the protection of LACCE against blue light needs to be further refined and studied using skin models.

## 4. Materials and Methods 

### 4.1. Analysis of the Compounds in LACCE Using UPLC-MS/MS

The compounds present in LACCE were qualitatively analyzed using UPLC-MS/MS. The LACCE was dissolved in water. The chromatography separation conditions were as follows: WATERS ACQUITY UPLC I-Class, ACQUITY UPLC BEH Amide chromatographic column, 1.7 μm, 50 × 2.1 mm; mobile phase, A: water (0.1% (*v*/*v*) ammonia), B: acetonitrile; column temperature, 40 °C; flow rate, 0.3 mL/min; injection volume, 0.5 μL; elution procedure, as described in Table 3; mass spectrum conditions, as described in Table 4; detection time, 20 min. The mass spectrometry conditions were as follows: WATERS Xevo TQD mass spectrometer, the carrier was argon, using the negative ion, MSE scanning mode; specific index requirements, capillary voltage 3.5 KV (ES-), cone-hole voltage 50 V.

### 4.2. Analysis of the Compounds in the LACCE Using HPLC

A method based on reversed-phase high-pressure liquid chromatography with a diode array detector (RP-HPLC-DAD; 1260 Infinity; Agilent Technologies, Santa Clara, CA, USA) was developed to determine the contents of seven compounds in the LACCE simultaneously, including syringin, chlorogenic acid, cynarin, isochlorogenic acid A, asiatica, isoquercitrin, and isochlorogenic acid C.

The sample was separated on a ZoRBAx SB-C18 reverse column (4.6 mm × 250 mm, 5 μm; Agilent Technologies). The mobile phases were 0.1% (*v*/*v*) formic acid in water and methanol (A and B, respectively), with an injection volume of 20 µL, and a flow rate of 1.0 mL/min. The liquid-phase elution gradients are presented in Table 5.

### 4.3. Cell Culture and Materials

The HFF cells were purchased from the National Laboratory Cell Resource-sharing platform (NICR, Beijing, China). Dulbecco’s modified Eagle’s medium (DMEM), penicillin–streptomycin (PS), heat-inactivated fetal bovine serum (FBS), phosphate-buffered solution (PBS), and trypsin EDTA solution A (0.25% trypsin and 0.02% EDTA) were obtained from Gibco (Thermo Fisher Scientific, Waltham, MA, USA). The COL-I/MMP-1 ELISA kit was obtained from Biotechwell (Shanghai, China). The primary antibody for COL-I/MMP-1 was obtained from Abcam (Abcam, Cambridge, UK). The *L. alpinum* callus culture was provided by Ancelbio (Chongqing, China) Biotechnology Co., Ltd. (Chongqing, China).

### 4.4. LACCE Preparation

The *L. alpinum* callus powder (10 g) was weighed out in a three-necked flask, to which 1,3-butanediol with a volume fraction of 30% was added, according to the material-to-liquid ratio of 1:50 (g:mL). Thereafter, the flask was placed on an electric jacket, heated at 100 °C and refluxed for 2.5 h. The mixture was then cooled at room temperature before the supernatant was collected.

### 4.5. Cell Culture

The HFF cells were cultured in DMEM containing 15% (*v*/*v*) FBS and 1% (*v*/*v*) PS, at 37 °C and 5% CO_2_. The cultured cells were used in the experiments when they were in the logarithmic growth phase.

### 4.6. Cell Viability Assay

Cell viability was determined using the CCK-8 assay (Biotechwell, Shanghai, China). Briefly, the HFF cells in their logarithmic growth stage were seeded into 96-well plates at a density of 1 × 10^4^ cells/well, at a temperature of 37 °C and 5% CO_2_ for 12 h; three parallel wells were used for each group. Thereafter, the HFF cells were pretreated with different doses of blue light [4] (6, 12, 18, 24, and 36 J/cm^2^) or different concentrations of LACCE [25] (2, 5, 10, 15, 20, and 50 mg/mL). Following incubation at 5 % CO_2_ and 37 °C for 24 h, the CCK-8 solution was added to the cells, in the dark; this was conducted at a temperature of 37 °C for 1 h. Subsequently, the absorbance of the samples was measured at 450 nm, using a microplate reader (TECAN; Männedorf, Switzerland), to calculate the cell survival rate.

### 4.7. Measurement of COL-I and MMP-1 Levels with ELISA

The HFF cells in their logarithmic growth stage were seeded into 6-well plates at a density of 3 × 10^5^ cells/well, 2 mL per well, 37 °C, and 5% CO_2_ for 12 h. Thereafter, the blank, blue light model, and sample groups were set up as follows: 2 mL of serum-free medium were added to the blank and blue light model groups; in the sample group, 2 mL of medium containing the various concentrations of LACCE (5, 10, 15 mg/mL) were added, and incubated at 37 °C and 5% CO_2_ for 6 h, while blue light irradiation was carried out. Following this, 2 mL of serum-free medium were added to each well, and cultured in an incubator at 37 °C and 5% CO_2_ for 18 h. Cell COL-I and MMP-1 levels were determined using a COL-I/MMP-1 ELISA kit (WELLBI, Shanghai, China) in accordance with the manufacturer’s instructions.

### 4.8. Measurement of COL-I and MMP-1 Levels with WB

The cell culture, addition of LACCE, and blue light radiation were carried out as described in Section 4.5. Following incubation with the blue light radiation device, the cells were collected and the total protein was extracted using a total protein lysis solution; thereafter, the cells were centrifuged at 12,000× *g* for 10 min. The supernatant was then collected and a BCA protein detection kit was used to determine the protein concentration. Protein denaturation was completed by preheating the metal bath at 100 °C for 5 min. The protein samples were separated using 10% SDS-PAGE electrophoresis and thereafter transferred to the PVDF membrane. The membrane was closed with 5% BSA, at room temperature, for 1 h. After TBST washing three times, the membrane was incubated with β-actin and the COL-I/MMP-1 primary antibody (Abcam, Cambridge, UK), overnight, at 4 °C. After washing three times again, the membrane was incubated with a secondary antibody at room temperature for 1 h. Finally, the protein bands were observed using ECL reagents and a chemiluminescent detection system.

### 4.9. Using Flow Cytometry for ROS Production Analysis

The cells were cleaned with PBS and 1 mL of DCFH-DA (10 μM) was added to them; thereafter, they were incubated in a 37 °C incubator for 30 min in the dark. Then, the DCFH-DA was discarded, and the cells were cleaned with PBS once again and prepared into a 1 × 10^5^ cells/mL cell solution. The cells were detected using the flow cytometer FL2 channel.

### 4.10. Using Flow Cytometry for Ca^2+^ Production Analysis

A 10 μL pre-configured Fluo-4 AM (4 μM) working solution was added to the cells, which were then incubated at 37 °C for 20 min; thereafter, the Fluo-4 AM working solution was removed, the cells were washed three times in the same way as described above, and the AM was cultured at 37 °C for 10 min to ensure complete de-esterification in the cells. Then, 1 mL PBS was added to the cells to prepare a 1 × 10^5^ cells/mL cell solution. The cells in the solution were detected using the flow cytometer FL2 channel.

### 4.11. Statistical Analysis

All data are expressed as mean ± standard deviation (SD) of three independent experiments. The normality test was conducted in this study, and the data meet the assumption of homogeneity of variance and normal distribution. Statistical significance was evaluated using a one-way analysis of variance (ANOVA), followed by Tukey’s test with the IBM SPSS Statistics 26.0 software (IBM, Armonk, NY, USA). Statistical significance was considered when the *p*-value was lower than 0.05. 

## 5. Conclusions

In this study, it was found that the LACCE promoted the production of COL-I, inhibited the secretion of MMP-1, and reduced blue light damage. It may play a role in inhibiting the activation of the OPN3-calcium pathway by blue light and reducing ROS expression. The compounds in the LACCE included leontopodic acid A and B, syringin, chlorogenic acid, cynarin, isochlorogenic acid A, asiatica, isoquercitrin, and isochlorogenic acid C. These results indicate that the LACCE has an anti-blue-light-damage effect on blue-light-induced fibroblast models, which further expands its application scope and provides theoretical support for the development of new raw materials in the natural food, medicine, and skin care industries. In addition, we need to further verify the influence of blue light on skin in this study through animal model tests and human tests in the future.

## Figures and Tables

**Figure 1 molecules-28-02172-f001:**
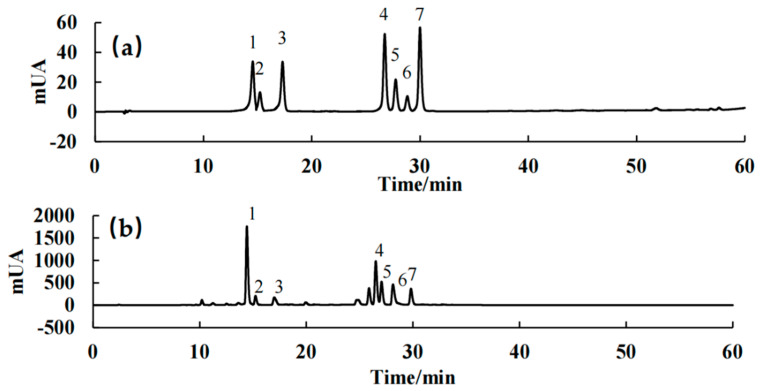
High-performance liquid chromatography (HPLC) profiles of the mixed standards (**a**) and LACCE (**b**). The numbers in the profiles represent the following: 1. Syringin, 2. Chlorogenic acid, 3. Cynarin, 4. Isochlorogenic acid A, 5. Asiatica, 6. Isoquercitrin, and 7. Isochlorogenic acid C.

**Figure 2 molecules-28-02172-f002:**
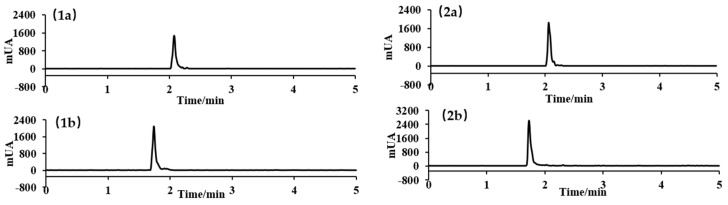
Ultra-performance liquid chromatography–tandem mass spectrometry (UPLC-MS/MS) profiles of the standards (**1**) and LACCE (**2**). The letters in the profiles represent the following: (**a**) leontopodic acid A, (**b**) leontopodic acid B.

**Figure 3 molecules-28-02172-f003:**
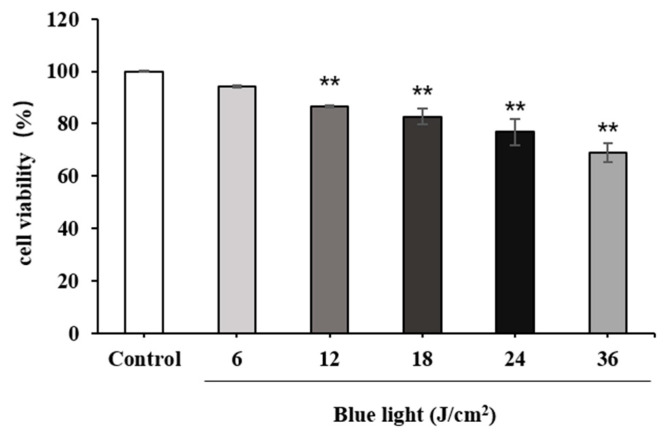
Effects of different doses of blue light on human foreskin fibroblast (HFF) cell viability. The data are expressed as mean ± standard deviation (SD) of three independent experiments (*n* = 3). The data were analyzed using one-way analysis of variance (ANOVA) followed by Tukey’s test. ** *p* < 0.05, versus cells without blue light treatment.

**Figure 4 molecules-28-02172-f004:**
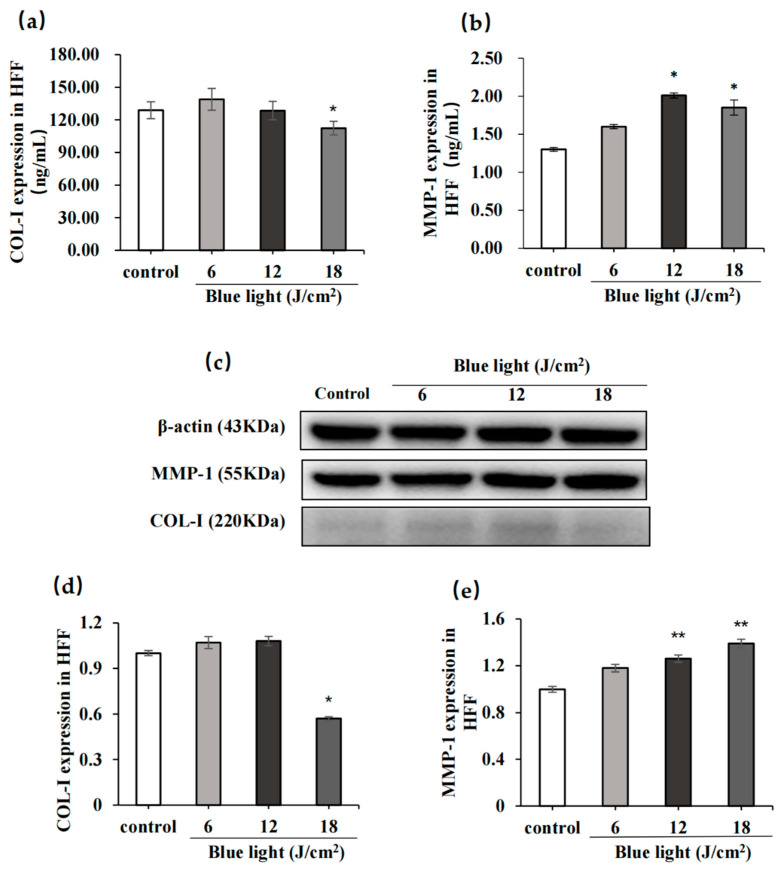
Results of the enzyme-linked immunosorbent assay (ELISA) and Western blotting (WB) experiments on HFF cells. (**a**,**b**) ELISA results after detecting collagen (COL-I) and metalloproteinase 1 (MMP-1) secretion levels in HFF cells under three blue light doses. (**c**–**e**) Level of COL-I and MMP-1 secretions in HFF cells under three blue light doses, from a WB assay. The data are expressed as mean ± standard deviation (SD) of three independent experiments (*n* = 3). The data were analyzed using one-way analysis of variance (ANOVA) followed by Tukey’s test. * *p* < 0.05; ** *p* < 0.01, vs. control group.

**Figure 5 molecules-28-02172-f005:**
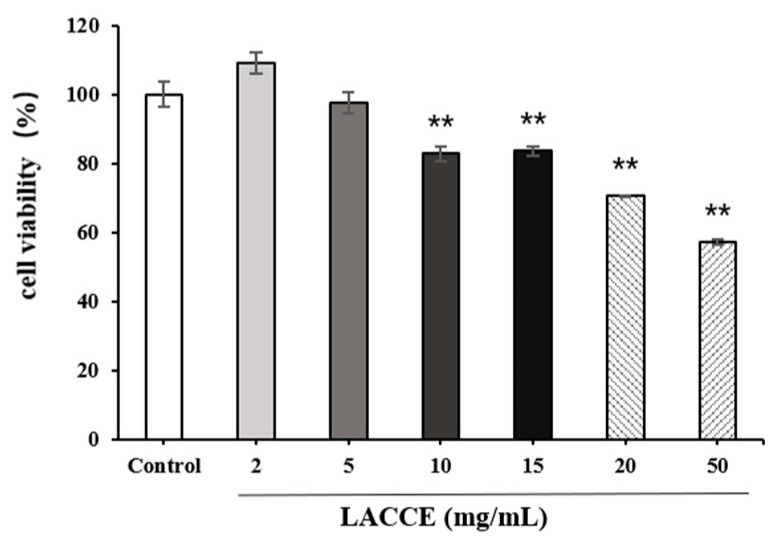
Effects of different concentrations of *Leontopodium alpinum* callus culture extract (LACCE) on HFF cell viability. The data are expressed as mean ± standard deviation (SD) of three independent experiments (*n* = 3). The data were analyzed using one-way analysis of variance (ANOVA) followed by Tukey’s test. ** *p* < 0.05, versus cells without LACCE treatment.

**Figure 6 molecules-28-02172-f006:**
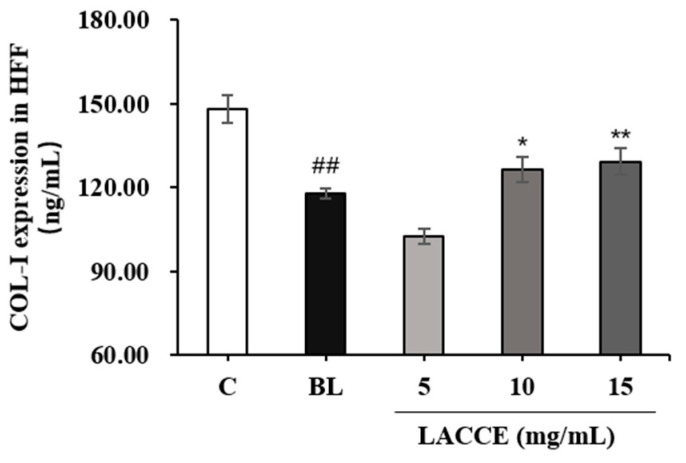
Effects of different concentrations of LACCE on COL-I. The data are expressed as mean ± standard deviation (SD) of three independent experiments (*n* = 3). The data were analyzed using one-way analysis of variance (ANOVA) followed by Tukey’s test. ^##^
*p* < 0.01 vs. control group (C). * *p* < 0.05 and ** *p* < 0.01 vs. blue light model group (BL).

**Figure 7 molecules-28-02172-f007:**
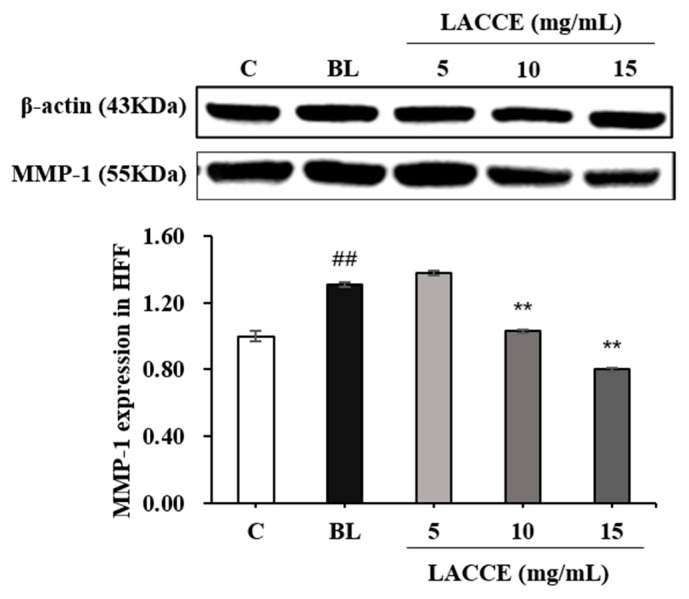
Effects of different concentrations of the LACCE on MMP-1. The data are expressed as mean ± standard deviation (SD) of three independent experiments (*n* = 3). The data were analyzed using one-way analysis of variance (ANOVA) followed by Tukey’s test. ^##^
*p* < 0.01 vs. control group (C). ** *p* < 0.01 vs. blue light model group (BL).

**Figure 8 molecules-28-02172-f008:**
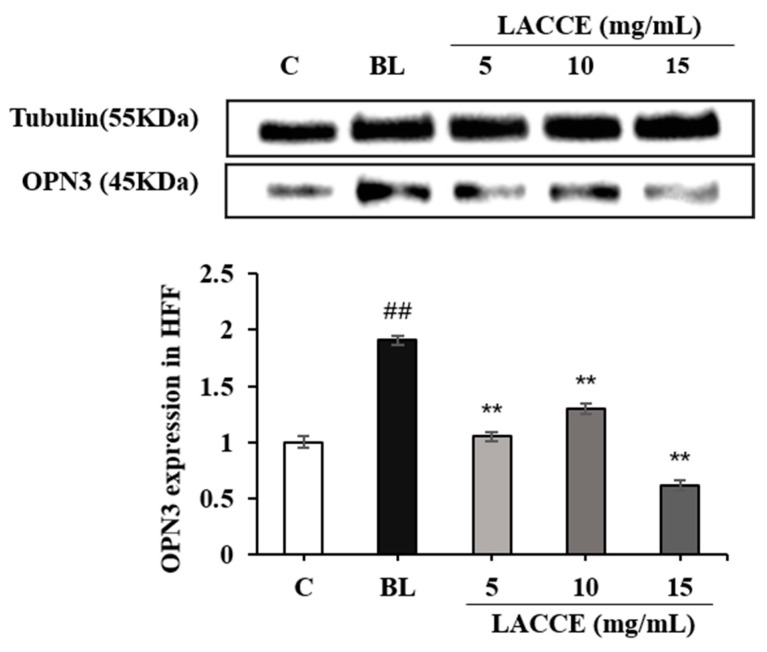
Effects of different concentrations of LACCE on OPN3. The data are expressed as mean ± standard deviation (SD) of three independent experiments (*n* = 3). The data were analyzed using one-way analysis of variance (ANOVA) followed by Tukey’s test. ^##^
*p* < 0.01 vs. control group (C). ** *p* < 0.01 vs. blue light model group (BL).

**Figure 9 molecules-28-02172-f009:**
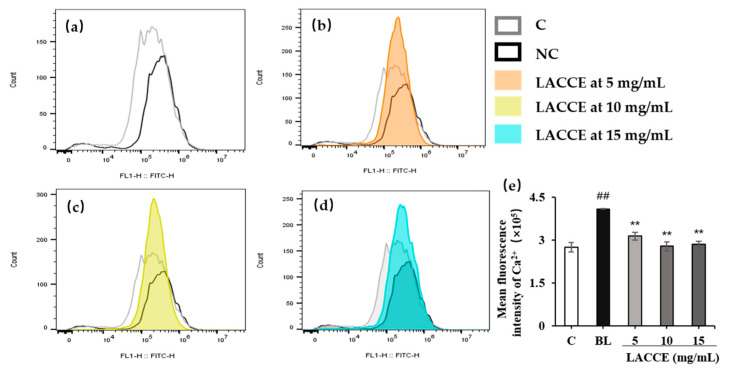
Effects of the LACCE on Ca^2+^. (**a**) Comparison of the Ca^2+^ inflow between the blank control group (C) and blue light model group (BL). (**b**) Influence of 5 mg/mL LACCE on Ca^2+^ inflow. (**c**) Influence of 10 mg/mL LACCE on Ca^2+^ inflow. (**d**) Influence of 15 mg/mL LACCE on Ca^2+^ inflow. (**e**) Quantitative analysis of the fluorescence intensity of Ca^2+^ (*n* = 3). The data are expressed as mean ± standard deviation (SD) of three independent experiments. The data were analyzed using one-way analysis of variance (ANOVA) followed by Tukey’s test. ^##^
*p* < 0.01 vs. C. ** *p* < 0.01 vs. BL.

**Figure 10 molecules-28-02172-f010:**
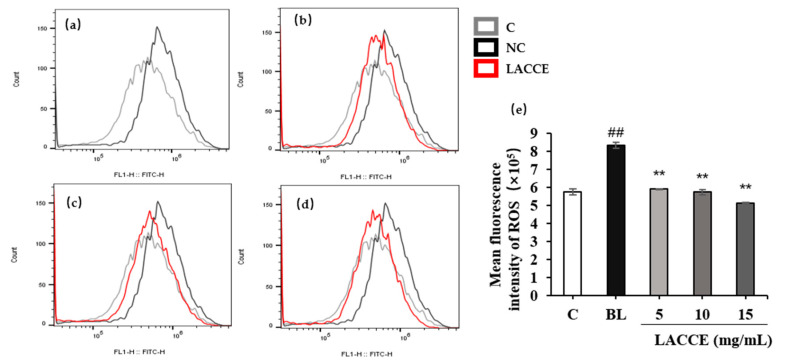
Effect of LACCE on ROS level. (**a**) Comparison of ROS levels between the blank control group (C) and blue light model group (BL). (**b**) Influence of 5 mg/mL LACCE on ROS levels. (**c**) Influence of 10 mg/mL LACCE on ROS levels. (**d**) Influence of 15 mg/mL LACCE on ROS levels. (**e**) Quantitative analysis of ROS fluorescence intensity (*n* = 3). The data are expressed as mean ± standard deviation (SD) of three independent experiments. The data were analyzed using one-way analysis of variance (ANOVA) followed by Tukey’s test. ^##^
*p* < 0.01 vs. C. ** *p* < 0.01 vs. BL.

**Table 1 molecules-28-02172-t001:** Quantitative linear ranges, correlation coefficients (R^2^), and quantitative analysis results of the standards.

No.	Components	Regression Equation	R2	Range (μg/mL)	Rt (Retention Times)	Composition (μg/mL)
1	Syringin	y = 18.73x − 9.335	0.9996	5–80	14.58	40.49 ± 2.43
2	Chlorogenic acid	y = 31.22x − 29.281	0.9999	5–80	15.28	125.96 ± 8.21
3	Cynarin	y = 25.48x − 16.612	0.9996	2–50	17.34	30.58 ± 3.40
4	Isochlorogenic acid	y = 30.74x − 41.030	0.9998	2–80	26.76	60.97 ± 5.85
5	Asiatica	y = 21.21x − 19.960	0.9999	2–80	27.68	52.33 ± 2.37
6	Isoquercitrin	y = 18.73x − 9.335	0.9997	5–80	28.89	128.27 ± 6.39
7	Isochlorogenic acid C	y = 18.73x − 9.335	0.9997	0.8–80	29.97	72.04 ± 2.96

**Table 2 molecules-28-02172-t002:** Calibration data and sensitivity of UPLC-MS/MS.

Parameter	Leontopodic acid A	Leontopodic acid B
Regression equation	y = 0.8106x − 5.2966	y = 1.5604x + 13.3110
Correlation coefficient (R2)	0.9962	0.9990
Range (ng/mL)	10–100	10–100
LOD (μg/mL)	3	0.4
LOQ (μg/mL)	10	5
Rt (retention times)	2.06	1.73
Composition (μg/mL)	184.13 ± 6.21	161.10 ± 3.85

**Table 3 molecules-28-02172-t003:** UPLC-MS/MS liquid phase condition.

Time/min	A %	B %	Flow Rate (mL/min)
Initial	0	100	0.3
1.00	0	100	0.3
1.50	70	30	0.3
4.00	70	30	0.3
4.01	0	100	0.3
5.00	0	100	0.3

**Table 4 molecules-28-02172-t004:** Mass spectrum condition.

Component	Parent Ion	Daughter Ion	Ion Source Mode	Capillary (kV)	Cone (V)	Collision (V)
Leontopodic Acid A	781.10	190.99	ES-	3.5	50	24
		295.01			50	24
		457.01			50	24
		617.01			50	24
Leontopodic Acid B	695.12	84.91	ES-	3.5	46	36
		208.99			46	36
		370.98			46	36

**Table 5 molecules-28-02172-t005:** HPLC detection method.

Time/min	A %	B %	Flow Rate (mL/min)
1	90	10	1
5	80	20	1
55	20	80	1
60	0	100	1

## Data Availability

Data are contained within the article.

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
