# Peer review of "Effects and Mechanism of the *Leontopodium alpinum* Callus Culture Extract on Blue Light Damage in Human Foreskin Fibroblasts"

_molecules, 2023, doi:10.3390/molecules28052172_

Round 1

Reviewer 1 Report

The present study brings relevant data for the development of better cosmetics which are able to protect against blue light, being relevant for the pharmaceutical and dermatological fields since the molecular approach explored in this work. This specific work brings relevant content for the comprehension of different doses of blue light radiation in the skin. Besides the very nice research delimitation, some points need to be improved. 

The first point which is necessary to correct is regarding the relation of blue light doses and electronic devices. It is already well-known that the blue light intensity from electronic devices is too small to really be a problem or healthy concern. I suggest the authors discuss this point in the introduction. Blue light is important for products such as sunscreens since this is the principal emission of blue light which concerns skin health. I also think it is necessary to include some points regarding blue light effects in different skin types.

I strongly suggest that the authors include more references in this work, it seems you were more interested to show the results without a very important discussion regarding this specific topic. There are many studies providing relevant data suggesting the utilization of different botanical extracts for the protection of visible light from the sun.

Please see the follow references below:

Mann, T., Eggers, K., Rippke, F., Tesch, M., Buerger, A., Darvin, M. E., ... & Kolbe, L. (2020). High‐energy visible light at ambient doses and intensities induces oxidative stress of skin—Protective effects of the antioxidant and Nrf2 inducer Licochalcone A in vitro and in vivo. Photodermatology, photoimmunology & photomedicine36(2), 135-144.

Tsuchida, K., & Kobayashi, M. (2020). Oxidative stress in human facial skin observed by ultraweak photon emission imaging and its correlation with biophysical properties of skin. Scientific Reports10(1), 9626

Tsuchida, K., & Sakiyama, N. (2022). Blue light-induced lipid oxidation and the antioxidant property of hypotaurine: evaluation via measuring ultraweak photon emission. Photochemical & Photobiological Sciences, 1-12.

Albrecht, S., Jung, S., Müller, R., Lademann, J., Zuberbier, T., Zastrow, L., ... & Meinke, M. C. (2019). Skin type differences in solar‐simulated radiation‐induced oxidative stress. British Journal of Dermatology180(3), 597-603.

The methodological approach is fine and the results are well presented, my suggestion is to shorten the text which is a little boring to read in some points since the repetitive information and long paragraphs. I suggest that the authors explain a little more about the HFF cell damage mode, is this model better than a full skin model? Why do they decide for this specific text?

I suggest first showing the botanical extract characterization by HPLC and after that the effects of visible light in the skin and after the effective protection of the present botanical extract. If possible, divide this by topics to make the text more attractive to read.

According to the results observed in the present study, do the authors can extrapolate these findings to electronic devices? If yes, why? Is a study conducted with cells strong enough for this observation?

In the end of the discussion it is mandatory that the researchers include a sentence pointing to the weak parts of the present study and its limitations. I strongly suggest that the authors include a paragraph ponting that this study was conducted in cells, not in a full skin model. Skin is a much more complicated and complex system and the observed results may be not the same.

The conclusion section should be improved after the observations presented by the reviewers. Is the conclusion section in the right position? I was a little lost regarding this.

Reviewer 2 Report

In this manuscript, several concerns need to be addressed, and extensive organization is required as follows:

1. The abstract should contain more information about the experimental protocol, particularly the tested levels of the extract and estimated parameters.

2. There is a problem with using abbreviations throughout the manuscript. The full term should be mentioned first with the abbreviation between paresis then the abbreviations should be exclusively used throughout the manuscript. E.g., Line 17: western blotting (WB), as the full term has not been repeated again, there is no need to give it an abbreviation. Also, Line 38: MMP1 should be presented as matrix metalloproteinase (MMP-1) then the abbreviation should be used further. Such errors have been repeated for many abbreviations throughout the manuscript.

3. Many formatting, typing, and grammatical errors exist. The manuscript is of a high need to English editing by a native speaker focusing on the following points:

- The writing style should be formal from the third-person perspective. Do not use we or our (E.g. line 381, we found).

- It is not preferable to begin sentences with abbreviations like OPN in lines 43 and 49.

Capitalization of letters. E.g. lines 14-15, Culture Extract should be culture extract.

4. Keywords: remove (LACCE). Replace OPN3; COL-I; MMP-1; ROS with the full term.

5. Introduction:

Line 36: “Current studies have shown [4]” should be “Previous studies,” and more than one reference should be cited.

Table 1 is redundant and should be deleted, and a paragraph containing its information should be added with references.

Has the information in lines 60-68 belong to the study of Yinghua[10]? If yes, clarify. If no, add the relevant references.

Lines 69-70 are the hypothesis of the study. Thus, it should be transferred to the end of the introduction before the aim.

Lines 73-75: add the reference.

Lines 75-77: add the references for these pharmacological effects 

6. Results:

Line 95-100: have the reduction in viability significant or not? Please, clarify.

Line 116: The cell survival rate decreased with increasing concentrations of the LACCE. It is very strange to find this reduction in cell viability with this plant extract. What are the attributions of the authors for this reduction?

The authors should mention how much change was induced by blue light in the results compared with the control group (i.e., % control). In addition, the degree of improvement in clove oil-treated groups compared to cadmium-exposed groups.

- In the figure legends, the authors should clarify the number of replicates. Add the full term of the abbreviation used in the legends. Clarify if the data has been presented as means±SE or SD.

Lines 130-132, 141-147, 156-160, 169-172, and 189-195: transfer to the discussion.

It is highly recommended to abbreviate the blue light group as “BL” instead of “NC”.

It is highly recommended to transfer the subheading 2.5. Analysis of the LACCE Chemical Composition to be at the beginning of the results and be “2.1”.

7. The discussion section is a major drawback of the manuscript. In its present form, it is just a repletion of the introduction section) lines 212-217; 227-235) with mentioning of the results again. The authors should rewrite this section and give a detailed discussion of their findings, and relate their findings to the detected bioactive of the extract with the possible interpretations and comparisons with the earlier studies.

8. Material and methods:

- On what basis have chosen the tested concentration range of both blue light and the extract?

- The reference to the protocols used in all methods is missed.

- transfer the subheading 4.9. 4.9. Analysis of the Compounds in LACCE Using UPLC-MS/MS to be the beginning of the material and methods section.

- Statistical analysis:

The experiment tests increasing levels of the extract. It is preferable to determine the linear and quadratic variable response to increasing levels of extract.

Does data meet the assumption of homogeneity of variances and normal distribution? Clarify if the authors run a homogeneity or normality test.

Clarify in what parameters T-test is used and in what parameters the One-way ANOVA test is used. What is the posthoc test used?

Reviewer 3 Report

It Is a well written article with good organized research procedure. Although, it is not considered a very attractive from the novel point of view, it is a very good scientific work. The results and methods are very clear and sufficient. Only the analytical process, although it is not a validated one, could be better and needs some minor improvements.

Figure 9. (b):  the 4th and 5th tops need better resolution. ‘

Nowhere in the paper the Rt (retention times) are referred, which is very important in analytical process.

Table 5, 6:  The number of  calibration curves (repeatability) is not written. The regression equation has to be the average of at least 3 curves. Please clarify.

Conclusion: You have to emphasize that further in vivo study has to be perform in the future, to confirm the in vitro results.

Round 2

Reviewer 1 Report

The present statement did not agree with the most modern literature regarding blue light: 

The exposure to blue light needs to be taken into account when extrapolating these findings to electronic devices. Previous studies have shown that blue light is damaging to human skin, so we need to further verify the effect of blue light on skin in this study through animal model tests and human tests.

The literature I have sent to the authors already shows that the intensity of blue light from electronic devices is not harmful for skin as the sunlight. This is basic physics. The amount of energy which a computer produces is not comparable with sunlight. Protection from blue light is a concern for sunscreens and not for cosmetic formulations to be utilized in front of computers. Cumulative effects can not be expect since the skin is an organ which is able to protect the increase of oxidative stress until certain point. Since the authors seems to continue to denied the most modern - and with stronger skin models. If the authors could prove that computers ad cellphones are harmful for skin in strong studies already published in the literature using better models and which are able to refute the other studies already presented in the first review, please include in the discussion, in the way you have presented seems to me that the scientific sounding of the presenting study is reduced.

Author Response

Dear Editor and Reviewers :
Thank you for taking time out of your busy schedule to review the manuscript. Now we have carefully corrected the manuscript for this revision. we have addressed the comments raised by the reviewers, and the amendments are highlighted in red in the revised manuscript. The revision instructions are as follows:
Question 1: The present statement did not agree with the most modern literature regarding blue light:
The exposure to blue light needs to be taken into account when extrapolating these findings to electronic devices. Previous studies have shown that blue light is damaging to human skin, so we need to further verify the effect of blue light on skin in this study through animal model tests and human tests.
The literature I have sent to the authors already shows that the intensity of blue light from electronic devices is not harmful for skin as the sunlight. This is basic physics. The amount of energy which a computer produces is not comparable with sunlight. Protection from blue light is a concern for sunscreens and not for cosmetic formulations to be utilized in front of computers. Cumulative effects can not be expect since the skin is an organ which is able to protect the increase of oxidative stress until certain point. Since the authors seems to continue to denied the most modern - and with stronger skin models. If the authors could prove that computers ad cellphones are harmful for skin in strong studies already published in the literature using better models and which are able to refute the other studies already presented in the first review, please include in the discussion, in the way you have presented seems to me that the scientific sounding of the presenting study is reduced.
Answer 1: Thank you for underlining this deficiency. I was wrong in my earlier interpretation of this question. The results observed in this study are not strong enough to extrapolate these findings to electronic devices.
Austin's study[1] showed that compared with the control group, ROS of iPhone 8+, iPhone 6 and iPad(first generation) white EDGL increased significantly by 81.71%, 85.79% and 92.98%, respectively, after 1 hour, indicating that fibroblasts would increase ROS production under the influence of electronic devices. But there was a non-significant change in apoptosis following irradiation with an iPhone 8+, iPhone 6, and iPad. And the biological effects of prolonged or repeated exposure were not fully understood. There is also evidence in the literature[2] that the intensity of blue light emitted by electronic devices is not as harmful to skin as sunlight. Therefore, further studies taking into consideration more parameters apart from the wavelengths, such as total received doses, irradiance and chronic exposure times, as well as skin type, are necessary to evaluate the real accumulative damage to skin.
1. Austin, E.; Huang, A.; Adar, T.; Wang, E.; Jagdeo, J. Electronic device generated light increases reactive oxygen species in human fibroblasts. Lasers Surg Med 2018, doi:10.1002/lsm.22794.
2. de Gálvez, E.N.; Aguilera, J.; Solis, A.; de Gálvez, M.V.; de Andrés, J.R.; Herrera-Ceballos, E.; Gago-Calderon, A. The potential role of UV and blue
light from the sun, artificial lighting, and electronic devices in melanogenesis and oxidative stress. Journal of photochemistry and photobiology. B, Biology 2022, 228, 112405, doi:10.1016/j.jphotobiol.2022.112405.
Thanks again to the reviewers and editors for your hard work! Please do not hesitate to contact us if there are any question. Best wishes to you!
Author: Xianyao MengXianyao Meng
2023.02.18

Reviewer 2 Report

The authors performed significant improvements in the manuscript. But, they gave a response to some comments in their response letter without making the required changes in the manuscript as follows:

1. The authors responded to the justification of the basis of choosing the tested concentration range of both blue light and the extract. But, this point has not been clarified in the method section. Also, the authors mentioned that the test range for LACCE concentration was set according to the amount added to the skin care product formulation. Where is the reference for the earlier information?

2. Does data meet the assumption of homogeneity of variances and normal distribution? Clarify if the authors run a homogeneity or normality test. The authors have not clarified this point in the manuscript.

3. The authors' response to the comment about the recorded reduction in cell viability with this plant extract is contradictory. The authors mentioned that "LACCE is a mixed extract of 1, 3-butanediol with complex components, which will stimulate cells to some extent with the increase of concentration". Thus it is expected to find an increase in viability, not a decrease. Also, the authors should discuss this point in detail in the discussion section.

4. In the figure legends, the authors should clarify n=? data have been expressed as means±SD, not just mentioned in the statistical analysis section.

Author Response

Please see the attchment
